# *Crocus pallidus* (Iridaceae)—A Neglected Species for the Bulgarian Flora and Critical Taxon in the Balkans

**DOI:** 10.3390/plants11050686

**Published:** 2022-03-02

**Authors:** Tsvetanka Raycheva, Kiril Stoyanov, Samir Naimov, Elena Apostolova-Kuzova, Yulian Marinov

**Affiliations:** 1Department of Botany and Agrometeorology, Agricultural University—Plovdiv, 4000 Plovdiv, Bulgaria; raicheva.tzveta@gmail.com; 2Department of Plant Physiology and Molecular Biology, University of Plovdiv, 4000 Plovdiv, Bulgaria; samir.naimov@gmail.com (S.N.); elena.apost@gmail.com (E.A.-K.); 3Natural History Museum—Plovdiv, 4000 Plovdiv, Bulgaria; julianmarinov@abv.bg

**Keywords:** Iridaceae, Bulgaria, ITS region, *Crocus pallidus*, *Crocus thracicus*, *Nudiscapus*, anatomy, morphology

## Abstract

For a long time, the Balkan endemic species *Crocus pallidus* has been unconfirmed and neglected for the flora of Bulgaria. It has remained an uncertain species from the Balkans, often listed as a synonym of *C. weldenii*. The morphological resemblance to the albinistic forms of *C. chrysanthus* has led to incorrect identification in the past, resulting in uncertainty regarding the distribution of this species in Bulgaria. In this regard, a detailed morphological and anatomical study of Bulgarian natural populations was carried out. A phylogenetic comparison in the ITS region of two Bulgarian populations of *C. pallidus* with other related species was conducted, revealing the distinction of *C. pallidus* populations from the closely related *C. weldenii*. The recently described new endemic species from Turkey, *C. thracicus*, shows very similar morphological and anatomical characteristics to *C. pallidus* and logically continues the distribution range in Eastern Thrace, along the Black Sea coast. We have a basis for suggesting that it should be treated as a synonym of *C. pallidus*.

## 1. Introduction

The genus *Crocus* L., with a center of diversity on the Balkan Peninsula and Asia Minor [1], constitutes 235 species [2]. The number of species has increased due to the descriptions of the new species from this area in the last few years [3,4]. However, a taxonomic revision of the genus *Crocus* in Bulgarian flora has not been conducted in the last 50 years. After the last taxonomic treatment in Bulgaria, the genus included nine species [5]. *Crocus pallidus* Kitan. & Drenk. was reported as a new species, based on the revised specimens of *C. chrysanthus* Herb. var. *albidus* Maw, collected from the region of North Macedonia [6]. Later, this species persisted in the research of Kitanov et al. [7] and was included in the list of species for the flora of Dobrudja [8]. For more than 35 years, this species has been neglected in the Bulgarian floristic literature and identification keys [9]. However, *C. pallidus* was mentioned in a review for the Balkan endemics in Bulgaria [10]. On the basis of the publication of Kitanov et al. [7], the fourth edition of “Conspectus of the Bulgarian Vascular Flora” [11] included *C. pallidus*, with a distribution range along the Black Sea Coast (Northern) and Northeast Bulgaria. Kitanov and other authors collected the available deposited materials from Bulgarian sites during 1965–1972. The limited studies so far have caused a lack of morphological and anatomical characteristics for *C. pallidus* in the referent taxonomic sources. Various authors have perceived white-flowered crocuses in the Balkans with different taxonomic statuses. Pulević [12] considered that the specimens described by Kitanov & Drenkovsky represent *C. chrysanthus* var. *citrinus* Velen. *Crocus pallidus* was neglected by Mathew [13] and noted as a heterotypic synonym of *C. weldenii* Hoppe & Fürnrohr. This has probably been the reason for *C. pallidus* being listed as a synonym of *C. weldenii* in the system of WCSP [14].

*Crocus weldenii* is a taxon with controversial taxonomic status. It has been perceived as a subspecies of *C. biflorus* s.l. [13]. This taxon is Balkan-endemic and Illyrian–Adriatic-endemic, distributed from Trieste to the border of Albania, with general distribution in Slovenia, Croatia, Herzegovina, Montenegro, Serbia, and North Macedonia [15]. The northwestern border of the species’ range reaches the vicinity of Trieste in NE Italy [16,17,18].

According to Ranđelović et al. [19], both species are found in Serbia. *Crocus weldenii* inhabits the hilly and southern regions, while *C. pallidus* is a species from the mountainous and more northern areas. The taxonomical status of *C. pallidus* remains very unclear [2].

All members of the genus *Crocus* have shown exclusively high diversity in their chromosome numbers [20]. The chromosome number of *C. biflorus* sensu Mathew (1982) varies between 2*n* = 8 and 2*n* = 36 [21]. The changes in the chromosome number in the related species play an important role in the speciation [22,23]. This could be an important feature if the close related taxa are different according to taxonomic numbers. Nevertheless, the representatives of the group discussed here have different morphology and the same chromosome number. The cytological models in *Crocus* are complex, with extensive dysploidy, and not satisfactorily explained [13].

The white-flowering crocuses on the Balkan Peninsula and Adriatic coast [19] have not been covered by the recent molecular and phylogenetic studies [24,25]. Furthermore, no annotated ITS sequences of *C. pallidus* were found in the NCBI GenBank Nucleotide database [26].

In the protologue of the older taxonomic literature, the morphology of *C. pallidus* has been laconic and incompletely reflected [5,17], which also leads to uncertainty of its status. A species named *C. thracicus* with a similar but more detailed morphology was described in northwest Turkey [27].

The information about the species in the Bulgarian floristic literature is based on sporadic reports, without precise localities and current diagnostic data. The various assumptions about the taxon have led to confusion and neglect in the Bulgarian literature. Moreover, the contradictory taxonomic treatments of white-flowered Crocuses from the Balkans need further study of a more thorough sample.

In this regard, our study aimed to provide detailed information on natural Bulgarian populations of *C. pallidus*, using the traditional morphological method, leaf anatomy, and an analysis of the literature data. To clarify the phylogenetic position among the related taxa, we used ITS internal and external spacers as taxonomically useful phylogenetic markers in *Crocus* [28].

## 2. Results

### 2.1. Description Based on Bulgarian Materials

A spring synanthous geophyte is shown in Figure 1 with the following features: corm oval-suborbicular, 13.5–25 mm in diameter, with membranous-coriaceous tunic, yellowish-brown, with separating basal rings; Basal rings 1.2–2.5 mm thick (Figure 1G) with unclear, slightly visible irregular dentation; plant height 100–210 mm; leaves 4–6(–7), 0.8–2.3 mm wide; flowers 2–3(–4); perigone segments niveous-white or outer ones with sprayed pale-blue coloration (Figure 1A,C); rounded to slightly acuminate; 19–33 × 6–11 mm long and 15–26 × 5–12 mm wide; perigone tube 6–10 cm long, white or bluish–lilac. (Figure 2); throat glabrous, white; filaments 3.8–10 mm; anthers 8–12(16) mm long with blackish basal lobes, or exceptionally with blackish stripes on the internal side, along the thecas (Figure 1B), on both sides of the white connective; stigma orange-reddish, trilobate (stylodia 3–5 mm), usually shorter than or equal to anther length; Capsule ellipsoid 15–21 × 4.3–7.4 mm (Figure 1D); seeds oval, dark brown 2.3–3.4 × 1.5–2.1 mm.

A comparison with the known morphological features of the closely related species is provided in Table 1.

### 2.2. Distribution

The species is Balkan-endemic. The distribution area extends over the territory from Serbia (eastern and southeastern), through Kosovo, North Macedonia, Bulgaria, and probably the European part of Turkey. The Black Sea coast represents the eastern border (Figure 3).

### 2.3. Phenology and Habitats

The flowering period lasts from the end of January to the end of March. The population of *C. pallidus* near the village of Kamen Bryag (SOA 062791) inhabits the open grass communities in the margin of an oak grove, accompanied by individuals of *Paliurus spina-christi* Mill., *Ficaria verna* Huds., *Crocus chrysanthus* (Herb.) Herb., *Geranium mole* L., *Fragaria vesca* L., *Viola* sp. div., *Trifolium* sp. div., Poaceae sp. div., and tufts of mosses. The whole habitat is surrounded by agricultural land. Approximately 150–200 individuals represent the population. A dense sympatric population of *C. chrysanthus* was found inside the grove. On the margin of the grove, populations of both species are mixed, whereas hybrids may occur (Figure 4). The locality is under anthropogenic pressure due to agricultural work, the main road laying on the western border of the population, and proximity to Kamen Bryag.

The population in Pomorie Narrow (SOA 062791) exists in sparse oak forest with the same accompanying species and *Anemone blanda* Schott & Kotchy. The few observed individuals of *C. chrysanhus* were flowering earlier than *C. pallidus* flowering time. The anthropogenic pressure is also significant (the habitat is an illegal landfill for household waste).

Under high anthropogenic pressure is the locus classicus of *C. p.* f. *bulgaricus* Kitan. & Drenk. It is an oak grove, subjected to unregulated logging and pollution, situated on the main road near the city of Dobrich (SO 21305).

The population near the Rudnik suburb of Burgas (SOA 063064) is under relatively low anthropogenic pressure. It is located in an open habitat, protected by the shrubs of *Paliurus*, and accompanied by *C. chrysanthus*.

### 2.4. Leaf Anatomy

The anatomical features of *C. pallidus* from the evaluated populations are presented in Table 2.

The leaves in genus *Crocus* have a unique bifacial profile of the cross-section, with central rectangular or square keel, and two lateral arms.

In the investigated populations, the keel is square to rectangular, with a wide base, and with a wide white stripe (15–30% of the leaf width). The lateral zones are convex due to the developed mesophyll around the bigger vascular bundles (Figure 5).

The arms’ length and their curving degree vary in the investigated populations. The ribs are present on the abaxial side of the arms (Figure 6B,E). Papillae are located in the apical regions of the arms and along the ribs (Figure 6A).

The adaxial epidermal cells are rectangular, with a thicker cuticle. The abaxial epidermal cells are elliptic in shape, while the cuticle layer is thinner. The stomata are located on the abaxial side of the leaves, (Figure 6E,G) in the zones of the arms and keel. The stomatal type is anomocytic. Single stomata are also sporadically present on the abaxial epidermis (Figure 6C,F).

The parenchyma in the central zone of the keel is represented by big rectangular cells with thin walls and without chloroplasts, forming a lacuna zone, visible like a white stripe along the leaf blade. The assimilating mesophyll in the zone of the arms consists of palisade and spongy layers. The palisade cells are oblong, organized in two rows, without any space between them. The spongy cells form zones of 2–4 rows near the abaxial epidermis, between the ribs. They have an elliptic to irregular shape, with slightly visible intercellular spaces between them (Figure 6C,E). Crystalline formations as oxalate sand or crystals with an approximately cubic shape are present in all three studied populations of *C. pallidus* (Figure 6D).

The vascular bundles are collateral, located in a row along the arms and the keel zone. The vascular tissues take 7–40(–61)% while the remaining bundles (about 2/3) consist of sclerenchymatous “caps”. The subterminal bundles in the folding area of the arms are bigger than the other (81–187 μm × 56–141 μm). The terminal bundles in the keel are the biggest in that area (89–190 μm × 70–136 μm). The number and size of the other bundles vary depending on the width of the section. The population from Kamen Bryag (SOA 062791) shows a bigger number of vascular bundles. Despite the quantitative variation in their number, in all of the sections, the biggest vascular bundle is subterminal, followed by 1–3 narrower terminal bundles in the shoulder.

### 2.5. ITS Sequences

On the basis of the ITS sequences, the phylogenetic tree (Figure 7) shows the known grouping of the species in section *Nudiscapus* B. Mathew. Two separated branches contain the series *Biflori* and the series *Reticulati*. The branch of series *Biflori* contains the Bulgarian samples of *C. pallidus* (in bold blue in the tree) together with *C. minutus*, *C. leucostylosus*, *C. ranjeloviciorum*, *C. punctatus*, *C. alexandrii*, *C. weldenii*, *C. pulchricolor*, and *C. adamioides*. A highly divergent subgroup contains the species from the Aegean islands and Italy (*C. biflorus*, *C. babadagnensis*, and *C. rhodensis*). In a divergent position remains a group of specimens determined as *C. chrysanthus*.

## 3. Discussion

*Crocus pallidus* belongs to subgenus *Crocus*, section *Nudiscapus*, series *Biflori* [13]. Due to the taxonomic changes and the growing number of the new species described, especially in the critical and volumetric section *Nudiscapus*, more detailed morphological descriptions are necessary for identification [29]. A disadvantage in the descriptions of the crocuses by earlier authors is the lack of detailed and comprehensive diagnostic descriptions. In the diagnosis by Kitanov and Drenkovski [6], as well as in the description by Ranđelović et al. [17], we did not find the data about the color of the anther’s basal lobes, or about the color variation in the outer perigone segments in *C. pallidus*. On the other hand, in the herbarium materials, deposited by Kitanov (SO 69556, 32784, 32795), the blackish lobes of the anthers are distinctly visible, although some of the samples are damaged and unsuitable for analysis (SO 13220; 103010). We suspect that this feature was overlooked by the authors because no other white-flowering *Crocus* species are known for the flora of Bulgaria.

Kitanov and Drenkovski [6] reported the two localities of *C. pallidus* in Bulgaria. According to a revision of the herbarium material from northeast Bulgaria of *C. chrysanthus* (SO 21305), the authors published a new form *C. pallidus* f. *bulgaricus*. The described variability is based on the relation between the leaf and the flower size. *Crocus p.* f. *pallidus* from North Macedonia (locus classicus, SO 30393) and Serbia is characterized by the leaves shorter than flowers, while f. *bulgaricus* can be recognized by the longer leaves compared to flowers. This form can be found in the region of Balchik (SO 13220) and Dobrich (SO 21305). So far, a reasonable position has been expressed in the literature for *C. pallidus* f. *bulgaricus*. Since the species and all its infraspecific taxa are equated as synonyms of *C. weldenii*, we consider that it is necessary to specify the infraspecific variability on the basis of the collected material and the data from the relevant literature sources. The variation in the meristic features of the evaluated populations is related to specific ecological conditions of the habitats. In our opinion, this variability is not a sufficient basis for taxonomic ranking. The studied populations also reveal polymorphism in the domain of leaf anatomy (Figure 5). The leaves of the plants near Kamen Bryag have two indistinct ribs along the arms, as mentioned by Rukšāns [2]. This feature may be related to the hybridization process and high variability in this population. The leaves of the other two populations have no ribs in the arms as reported for *C. thracicus* [27].

Our study reveals the presence of nonfunctioning stomata on the abaxial epidermis in the two populations from the Black Sea coast (Figure 6F). This detail was not registered in the population from Roudnik. Data about the abaxial stomata in *Crocus* were not found in the relevant anatomical literature. Moreover, the available information about the leaf epidermal structures is pretty limited [28,30].

*Crocus chrysanthus* and *C. pallidus* are spring synanthous geophytes, with an overlapping flowering period. They are often found in sympatric populations, occasionally resulting in hybrids between them (Figure 4). Our morphological findings correspond to the descriptions of *C. pallidus* [2] and *C. thracicus* [27], with minor quantitative differences. For example, the corms in the population from Kamen Bryag are much bigger than the other examined populations and the data from the literature. This feature could be a consequence of the differences in the biotope (soil type, humidity). Very variable color of the outer perigon segments, from pale blue to white with a slightly visible yellow stripe, was noted in the population near Kamen Bryag. A probable reason for the variation is the overlapping flowering period with *C. chrysanthus*. We noticed the hybrid with an intermediate coloration of the perigone segments (white and yellow) in the same population (Figure 4B).

The existence of clear and mixed populations demonstrates that *C. pallidus* is a distinctly differentiated species. The spontaneous hybrid forms and the unclear morphological differentiation of the features between *C. pallidus* and *C. chrysanthus* s. l. are evidence of the close relationship and probably recent divergence, assuming the distribution area of both species. We consider that introgressive hybridization is a reason for the higher variability in the mixed population. Sympatric populations of *C. pallidus* with *C. crysanthus* with overlapping flowering time have already been mentioned [17]. A similar feature was stated for *C. thracicus* × *C. chrysanthus* [27]. Our investigation shows a morphological similarity between *C. pallidus* and *C. thracicus*. Since an ITS sequence of *C. thracicus* has not been deposited in the NCBI Nucleotide database, its phylogenetic relationship with Bulgarian populations of *C. pallidus* remains unknown. Geographically, populations of *C. thracicus* are located very close to the Bulgarian localities of *C. pallidus*, especially those in the south (Figure 3). Furthermore, the authors compare *C. thracicus* with the closely related *C. alexandri* Ničič & Velen. and *C. weldenii*, while *C. pallidus* is mentioned but neglected as a morphological description at the same time. According to Yüzbaşioğlu et al. [27], *C. pallidus* has a yellow throat. This qualitative feature is not correctly cited. In our study on four Bulgarian populations, as well as in the deposited herbarium specimens (see Section 4), the perigone throat is white. We found the same data about *C. pallidus* in the literature (Mathew, 1982; Rukšāns 2017) and in the evaluated herbarium specimens in SO. Kitanov and Drenkovsky [6] and Ranđelović et al. [17] did not describe the key features such as the throat color, as well as the dark basal lobes of the anthers, in *C. pallidus*. The examined specimens, including the deposited material by Kitanov and Drenkovsky, stably show the white throat and blackish basal lobes of the anthers. The same morphology has been noted in the description of *C. thracicus*. The reason for this assumption is also given by the close morphological and anatomical parameters of the two species, the habitat conditions, and the elevations (Table 1). In the white-flowering crocuses group of the north of Balkan Peninsula and southeastern Europe, the only member with a yellow throat is *Crocus malyi* Vis. (section *Crocus*, series *Versicolores*) with a distribution area on Velebit Mts. in Croatia, Bosna, and Herzegovina [2,31,32].

Molecular and phylogenetic methods and analyses are reliable tools for explaining the complex relationships between taxa in the nonmonophyletic section *Nudiscapus*, particularly the white-flowered species of the *Biflori* series distributed in the Balkans. The sequences of both populations of *C. pallidus* with Bulgarian origin are identical. The generated phylogenetic tree with the *C. adamii* s. str. group as the outgroup complements the data from Harpke et al. [28] without major changes in the topology. The samples of *C. pallidus* are situated in the group of the polymorphic series *Biflori*, together with *C. minutus* Kerndorff & Pasche [33] from Turkey (Lycia), *C. punctatus* Kerndorff, Pasche & Harpke [24] from southern Turkey (Isparta and Burdur), and *C. leucostylosus* Kerndorff, Pasche & Harpke in [24], with a native range Western Turkey (Denizli), *C. alexandri*, *C. weldenii*, and *C. randjeloviciorum*. The big clade in ser. *Biflori* contains species with discrete anatomical and morphological differences; however, according to the ITS similarity, they are genetically closely related and remain in an unresolved phylogenetic position.

## 4. Materials and Methods

### 4.1. Examined Specimens

The samples were collected during terrain work in 2019–2021. We used the materials deposited in the national herbariums to compare some qualitative and metric features. The differences between this taxon with the related *C. thracicus* and *C. weldenii* were taken from protologues and photos of herbarium specimens and were reviewed in a table.

The examined specimens are listed below. The vouchers for the ITS sequences are signed with their NCBI numbers. New and unpublished chorological data are signed with an asterisk (*). The specimens are listed by country, floristic region, MGRS square, toponym, elevation, exact geographic coordinates (if present), date, and collector, followed by herbarium acronym and entry number.

***Crocus pallidus***:

**North Macedonia: 34TDM71**. Near Mavrovo. 19 March 1968 (coll. Drenkovski & Kitanov) SO 32794; **34TEM11**. „Selo Zrkle, Porec–dolina na Treska“, near the village of Zrkle, Treska Valley, 2 March 1972 (coll. B. Drenkovski sub *C. chrysanthus* Herb var. *albidus* G.Maw, rev. B. Kitanov, **Type**) SO 30393; **34TEM36**. Skopska Crna Gora, 600 m. 19 February 1972 (coll. Drenkovski sub *C. crysanthus* Herb. var. *albidus* G.Maw; rev. Kitan. et Drenk, **Isotype**) SO 30392, 32795; **BULGARIA:** * **Black Sea Coast (Southern)**: **35TNH52**. Pomorie Narrow–Pandarlak locality, N42.71911 E27.72632, 183 m, 27 February 2020 (coll. Raycheva & Stoyanov) SOA 062797 (NCBI record MW775330). **Black Sea Coast (Northern)**: **35TNH89**. “Zlatni Pyasatsi” Nature park, N43.303 E28.034214, 250 m, 13 March 2015 (coll. Y. Marinov) SOA 063074; **35TNJ90**. Near the town of Balchik, April 1965 (coll. Minchev & Kitanov sub *C. biflorus* Mill. var. *albus* Herb.; rev. *C. pallidus* Kitan. 20 Mar. 1973) SO 13220; **35TPJ21**. Kamen Bryag, 7 March 1973 (coll. Georgieva & Kitanov) SO 69557, 8 March 1973 (coll. Dimova & Kitanov) SO 69556; N43.45199 E28.54402, 41 m, 28 February 2020 (coll. Raycheva & Stoyanov) SOA 062791 (NCBI record MW775331). **Northeast Bulgaria**: 35TNJ62. Kabaklaka locality, near the city of Dobrich, May 1971 (coll. Dencheva; Type: *C. p.* f. *bulgaricus* Kitan. & Drenk.) SO 21305; **35TNJ70**. The village of Batovo, 1975 (coll.?) SO 103010; **35TNJ80**. Dezpelin locality, near the village of Topola, 59 m, N43.403085 E28.069111, 7 April 2020 (obs. Zh. Barzov) photo set; **35TPJ14**. Bezhanovo, 36–69 m, N43.71868 E28.41449, 16 February 2018 (obs. Zh. Barzov) photo set. * **Toundja Hilly Plain**: **35TNH42**. Noth of the Roudnik neighbourhood of Bourgas, 150 m, N42.638119 E27.489508, 27 February 2021 (coll. T. Raycheva & K. Stoyanov) SOA 063064.

***Crocus thracicus*** (photo scans):

**Turkey:** A1(E), Kırklareli: Vize, Saray-Vize yolu, c. 10 km, Quercus sp. & Paliurus spina-christi Mill. aҫıklıkları, 170 m, 8 February 2014 (coll. S. Yüzbaşioğlu, S. Aslan, İ. Sözen & F. Canız, **Holotype**–ISTE 102922; **Isotype**–DUOF 5630.

***Crocus weldenii*** (photo scans [34,35,36,37,38]):

**Croatia**: Istrien Kvarner/Carnaro/Quarner Bucht, okot Krk/isola di Veglia/Insel Vögls, NW Draga Bašćanska/Bescavalle, E-exponierter Hang zum Bach Suha Ričina/Fiumera, W über dem Wasserfall; 14°41′12,40″ E 45°01′09,61″ N, 174 m, 16 February 2019 (coll. Rottensteiner, Walter K. & Rottensteiner, Marica) GJO 0099926; Gemeinde Bakar; 30–300 m Seehöhe; 14°32′00″ E 45°18′25″ N, 4 February 1886 GJO 004853; mt. Marian, nan loin de Spalato, May 1830 (coll. Steudel) K 000802457; **ITALY**: Istria. In rupestribus ad Gabrovizza prope Prosecco (coll. Solla) O V 2187017; **cult**. L 1476872.

### 4.2. Anatomical Investigations

Quantitative measurements and observations of qualitative features of fresh samples for the morphological analysis were conducted. To examine the leaf anatomy, 10 individuals per population, from three populations in total, were collected during the flowering time and conserved in 75% ethanol. Transverse cross-sections and epidermal areas of leaves from each individual were manually constructed from the middle part. The epidermis was peeled using a surgical scalpel. The microscopic slides were fixed with glycerin. The snapshots of the microscope slides were taken using a Leica 750 digital microscope, while the measurements of the characters were completed using Micam 2.4 software [39]. The total of 24 anatomical features included section width and height, arm length, white stripe width, the number and size of vascular bundles, palisade tissue height, height and width of palisade and spongy cells, and the height and width of the adaxial and abaxial epidermal cells. Abaxial epidermal cells, including the size of the stomata, were measured in the area of the stomatal rows (between the ribs). The values were expressed through the basic statistic parameters (mean, standard deviation, maximum and minimum).

### 4.3. Molecular Methods

Three populations were evaluated by anatomical parameters. DNA, isolated from two of the populations, was used for sequencing of the small subunit ribosomal RNA gene, partial sequence (ITS 1, 5.8S ribosomal RNA gene, and ITS 2, complete sequence; and large subunit ribosomal RNA gene, partial sequence). The genomic DNA was isolated using a Dneasy Plant Mini Kit (QIAGEN). Briefly, 50 μg of plant material was ground in liquid nitrogen and processed according to the manufacturer’s requirements. DNA concentration and quality were determined spectrophotometrically at a wavelength of 260 nm using an Epoch microtiter plate reader and T3 plate protocol. The ITS region (ITS1, 5.8S rDNA, ITS2) was amplified using the following primers: ITS-A (5′–GGAAGGAGAAGTCGTAACAAGG–3′) and ITS-B (5’–CTTTTCCTCCGCTTATTGATATG–3′), as described in Tirel et al. [40]. The 50 μL reaction mixture contained 1× reaction buffer, 200 μM of dNTPs, 0.2 μM of each primer, 100 ng of genomic DNA, and one unit of Q5 High Fidelity DNA polymerase (New England Biolabs). The PCR thermal cycler steps were as follows: initial melting of the reaction mixtures at 94 °C for 45 s, followed by 30 cycles at 94 °C for 10 s for denaturation, 10 s at 62 °C for primer annealing, 30 s at 72 °C for primer extension, and a final elongation step of 2 min at 72 °C. Amplified PCR products were separated by 0.8% agarose gel electrophoresis, excided from the gel, and purified using a QIAquick Gel Extraction Kit (QIAGEN). The purified DNA fragments were subsequently sequenced with Microsynth Company (Balgach, Switzerland) technology. Chromatograms were corrected manually with DNAStar software (Lasergene, Madison, WI, USA). The sequences were uploaded to the NCBI Gene database under accession numbers MW775331 (voucher SOA 062791), MW775330 (voucher SOA 062797), and OM368594 (voucher SOA 062596).

### 4.4. Phylogenetic Analyses

The obtained nucleotide sequences were blasted against nucleotide sequences from the NCBI Nucleotide database [41]. The best hits were downloaded and used for phylogenetic analysis. The alignment of the sequences (Figure A1) was achieved using the ClustalW Multiple alignments [42]. The phylogenetic analysis was conducted using Bayesian phylogenetic inference with MrBayes 3.2 [43]. The parameters of the analysis were the same as described by Harpke et al. [28]: 2 × 4 chains for two million generations, nuclear data set ГTP + G + I, sampling tree per 1000 generations, two independent runs. The result was visualized as a tree using TreeGraph 2 [44]. The analysis included 35 nucleotide sequences, cited as numbers of NCBI entries in the phylogenetic tree (Figure 7, Table A1). We accepted the outgroup as the *C. adamii* s. str. group [28].

## 5. Conclusions

The results of the taxonomical, morphological, anatomical, and molecular studies of four Bulgarian populations of *C. pallidus* confirmed the presence of this species in the flora of Bulgaria. In this study, we provided detailed information about the leaf anatomy of *C. pallidus* from Bulgaria for the first time.

Our studies were specifically focused on obtaining ITS data for *C. pallidus* that have not been studied so far. The investigated ITS sequences of the two populations of *C. pallidus* were identical but different when compared to *C. weldenii*. The morphological differences between these two species were also noticeable. Consequently, *C. pallidus* should be treated as an autonomous species. On the other hand, *C. thracicus* shares similar morphological and anatomical features with *C. pallidus*. It is distributed close to the localities of *C. pallidus*. Both species have hybrids with *C. chrysanthus*. Summarizing the previous facts, there is a possibility that *C. thracicus* represents a taxon conspecific with *C. pallidus*.

According to recent data, the distribution area of *C. pallidus* consists of disjunct populations in Serbia–Macedonia and East Bulgaria–European Turkey.

We believe that this information will be helpful for future studies related to the taxonomy and phylogeny of the white-flowering *Crocus* species from the Balkans and neighbouring regions.

## Figures and Tables

**Figure 1 plants-11-00686-f001:**
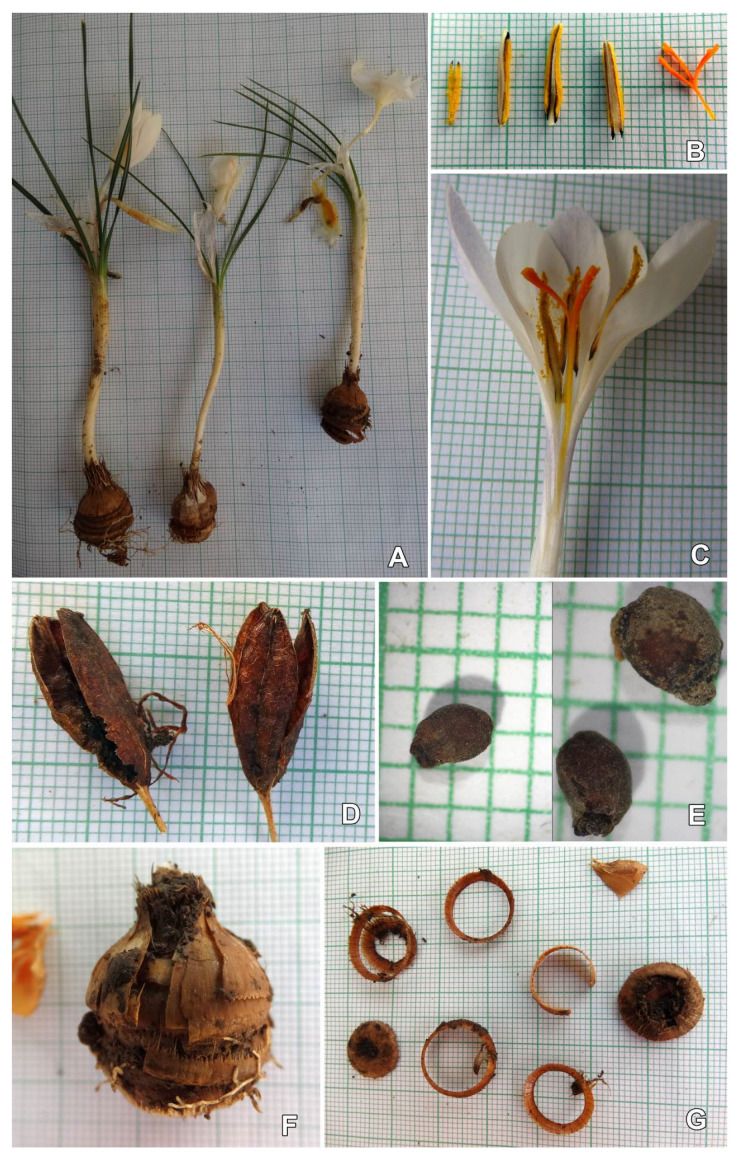
Morphological features of *Crocus pallidus* (specimen SOA 062791, grid 1 mm): (**A**) whole plant; (**Β**) anthers and stigma; (**C**) flower section; (**D**) mature capsules; (**E**) seeds; (**F**) corm; (**G**) basal rings and bottom of the corm.

**Figure 2 plants-11-00686-f002:**
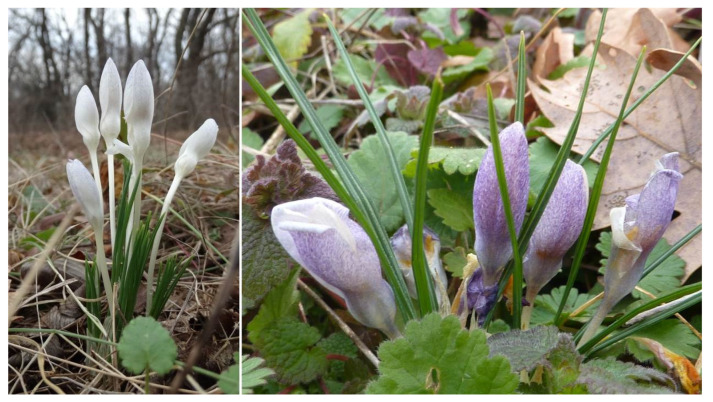
Variation in the color of the outer perigone segments of *Crocus pallidus*—white and speckled violet (population with voucher 062791).

**Figure 3 plants-11-00686-f003:**
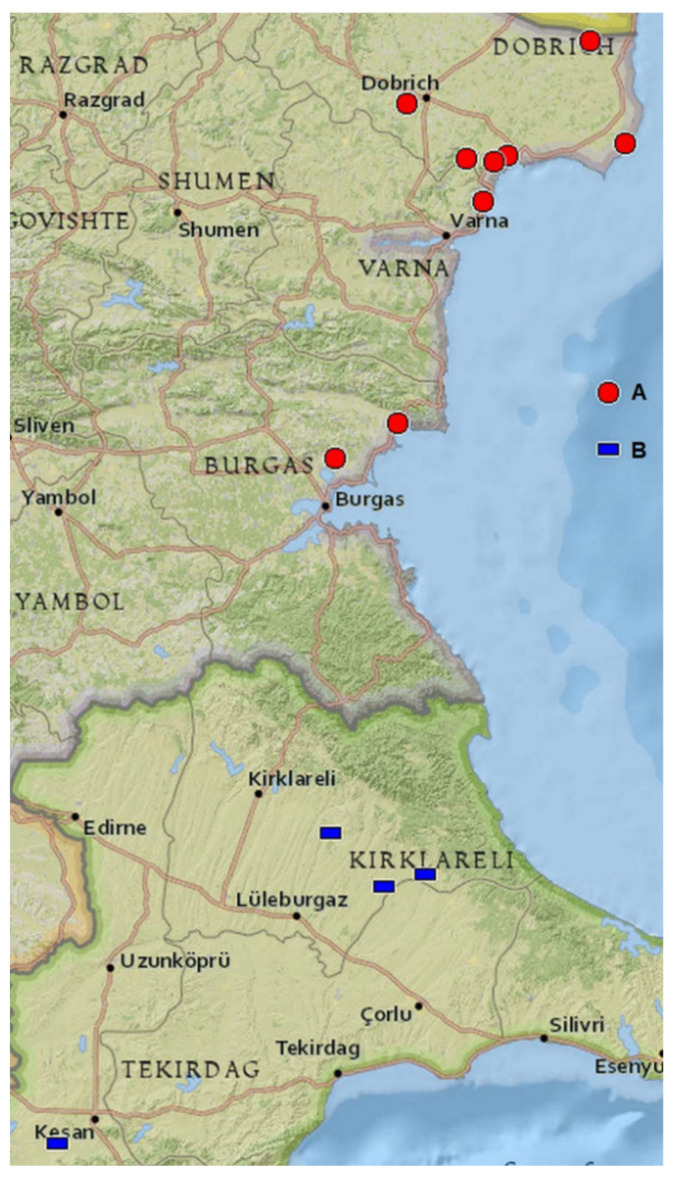
Localities of *Crocus pallidus* (**A**) in Bulgaria and *C. thracicus* (**B**) in Turkey. Map created at GPSVisualizer.com (accessed on 16 March 2021). Leaflet | NGS maps from ESRI/ArcGIS.

**Figure 4 plants-11-00686-f004:**
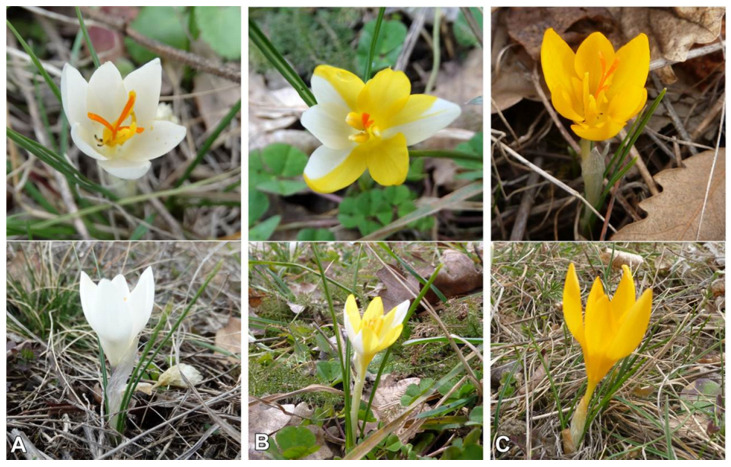
*Crocus pallidus* (**A**), *C. chrysanthus* (**C**), and their hybrid (**Β**) in the population with voucher 062791.

**Figure 5 plants-11-00686-f005:**
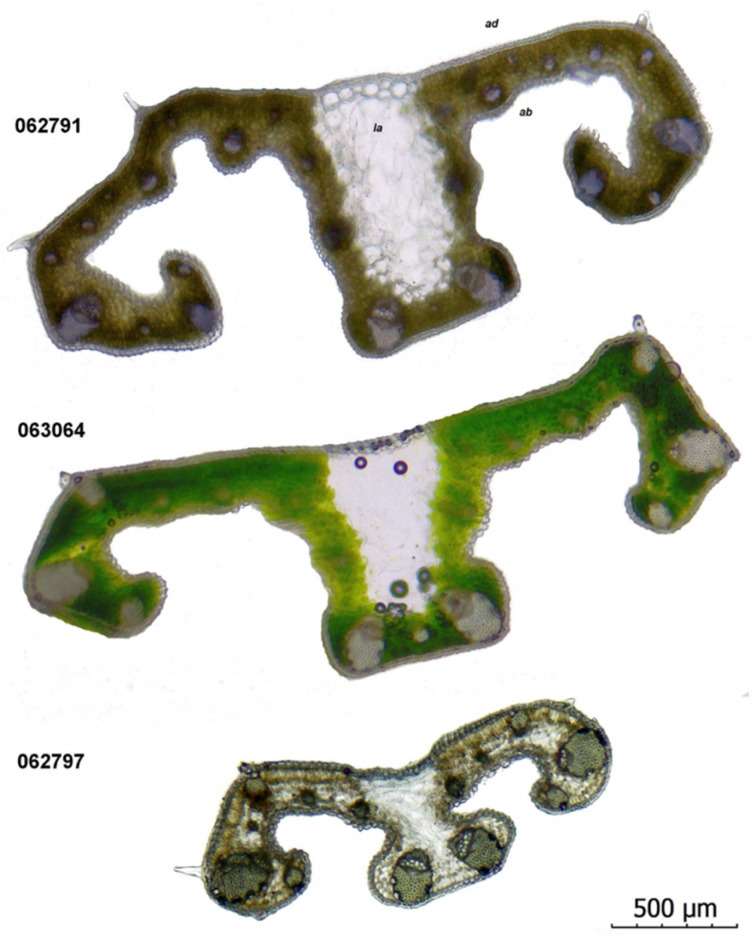
*Crocus pallidus* leaf cross-sections (4×). Abbreviations: ad—adaxial side, ab—abaxial side, la—lacuna area. Voucher numbers are shown near the cross-sections.

**Figure 6 plants-11-00686-f006:**
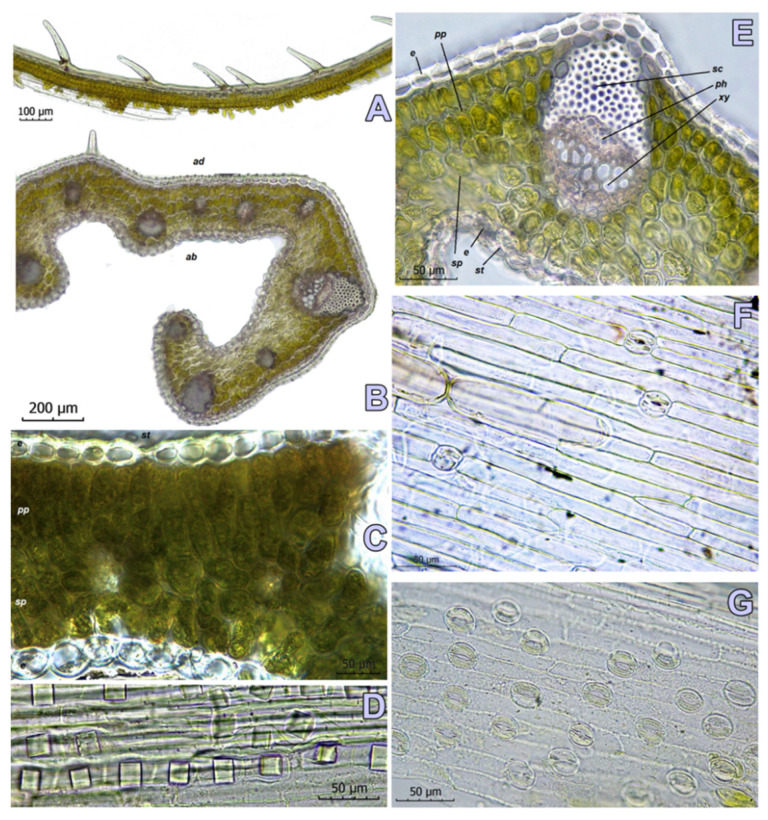
Leaf anatomy of *Crocus pallidus*. (**A**) Papilles on the ribs—adaxial surface; (**B**) section of the shoulder (SOA 062791); (**C**) detail of the leaf with visible adaxial stomata (SOA 062797); (**D**) crystals; (**E**) subterminal vascular bundle in the shoulder; (**F**) adaxial epidermis with stomata; (**G**) abaxial epidermis with stomata (SOA 062791). Abbreviations: ad—adaxial side, ab—abaxial side, la—lacuna area, e—epidermis, pp—palisade parenchyma, sp—spongy parenchyma, sc—sclerenchyma cap, ph—phloem, xy—xylem.

**Figure 7 plants-11-00686-f007:**
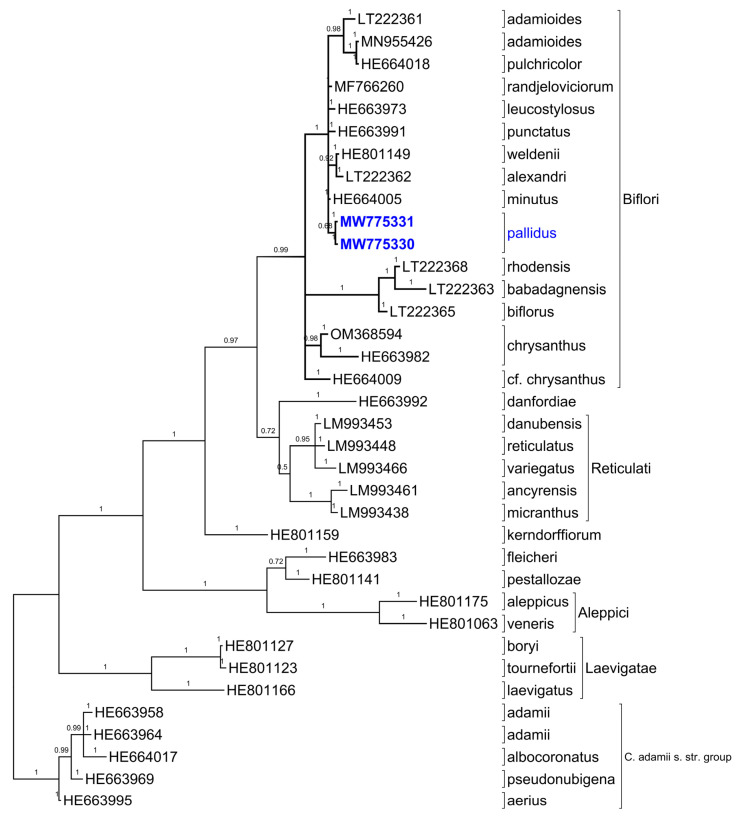
Placement of *Crocus pallidus* in the phylogenetic tree of sect. *Nudiscapus* obtained by Bayesian phylogenetic inference of the nuclear rDNA ITS regions using the methodology of Harpke et al. [28]. Posterior probabilities are designated by numbers. See Table A1 for details.

**Table 1 plants-11-00686-t001:** A comparison of morphological features of *Crocus pallidus* (current data), *C. weldenii* [13,17], and *C. thracicus* [27].

	*C. pallidus*	*C. weldenii*	*C thracicus*
Corm	13.5–25 mm	13–19.7 mm	10–12 mm
Plant height	(8)10–20 cm	8–17 cm	7.35–12 mm
Teeth of the basal rings	Irregular	Regular/Irregular	Irregular
Count of leaves	3–5	3–5	3–4(–5)
Leaf width	0.84–2.3 mm	max 1 mm	0.75–1.3 mm
Ribs on the abaxial leaf surface	Missing or vaguely visible	1–2, clearly visible	Missing
Color of the perigone tube	White, rarely violet	White, often violet	White, rarely violet near the apex
Color of outer perigone segments	White or sprayed in violet	White	White or sprayed in violet
Size of perigone segments (out/in)	19–33 × 6–11/15–26 × 5–12 mm	19–37 × 4.8–12.8	17–24 × 6–9/14.6–23 × 5.7–8.7 mm
Perigone throat	Glabrous, white	Yellow	Glabrous, white
Filaments	3.8–13.7 mm	8–16 mm	10–13.5 mm
Anthers	8–16 mm, yellow, with blackish basal lobes, or entirely black edge	yellow, without blackish lobes	7.3–11.8 mm, yellow, with blackish basal lobes.
Capsule	Ellipsoid; 15–21 mm long; 4.3–7.4 mm wide.		Ellipsoid, about 14 mm long
Seeds	2.2–3.4 × 1.5–2.1 mm, with convex caruncle		2.5 mm in diameter, with convex caruncle
Distribution	Bulgaria (Eastern parts), North Macedonia, Serbia.	Italy, Albania, Serbia.	Turkey in Europe (Thrakia)
Elevation	30–190 m	100–750 m	45–170 m
Flowering period	February–March		February–March

**Table 2 plants-11-00686-t002:** Anatomical parameters of the examined populations of *Crocus pallidus*. Values are given as a range (minimum–maximum), mean ± standard deviation.

Population *	062791 Kamen Bryag	062797 Pomorie Narrow	063064 Roudnik
Section width, µm	1701–2224 1927.6 ± 133	955–1722 1210.9 ± 193	910–2017 1432.6 ± 327.9
Section height, µm	709–807 761.1 ± 39.8	356–664 481.8 ± 89.2	480–633 562.8 ± 54.8
Arm width, µm	657–973 830.1 ± 96.3	385–696 547.2 ± 91.1	455–1071 703.4 ± 146.6
White stripe to leaf width ratio, %	15.1–21 17.4 ± 2	12.7–28.8 20 ± 4.2	14.9–19.3 16.64 ±1.5
Vascular bundles, count	23–25 23.8 ± 1	13–15 13.8 ± 1	15–17 15.5 ± 0.3
Vascular bundles, height, μm	25.9–177 81.7 ± 45.3	27.5–187 95.6 ± 45.8	33.2–190 104.6 ± 50.1
Vascular bundles, width, μm	17.9–103 59.2 ± 24.6	22.9–141 73.2 ±36.5	23.9–127 74 ± 29.8
Vascular tissues in the bundles, %	22.2–50.7 37.3 ± 8.2	7–61 19.9 ±9.3	4.1–30.8 19.4 ± 7.7
Palisade tissue: thickness, μm	13.7–77 44 ± 14.5	38–80 56.8 ± 8.8	25.1–75 57.9 ± 11.2
Spongy tissue-thickness, μm	25–91 48 ± 16.6	43–104 61.6 ± 13.8	33.8–65 46.3 ± 9.7
White stripe, µm	287–407 341 ± 49.2	150–295 241.1 ± 44.5	247–305 276.9 ± 22.8
Adaxial epidermal cell: length, µm	173–290 236 ± 47.3	98–431 237.2 ± 61.7	155–473 286.2 ± 66.8
Adaxial epidermal cell: height, µm	15–23.8 18.5 ± 1.96	11–22.6 17.83 ± 2.56	6.7–21 15.3 ± 3
Adaxial epidermal cell: width, µm	10.4–23.1 17.5 ± 2.93	11.5–20.8 16.45 ± 2.4	9.8–23.5 16.8 ± 3
Palisade cell: height, µm	13.8–36.4 26.4 ± 4.25	21.3–40.7 32.1 ± 6.49	17.2–41.3 30.3 ± 5.3
Palisade cell: width, µm	8.8–21.2 13.9 ± 2.47	9.6–18.6 13.53 ± 2.46	8.9–17.5 12.7 ± 2
Spongy cell: height, µm	14.4–29.6 21.4 ± 3.14	19.1–25.1 22.36 ± 2.07	13.9–28.6 20.7 ± 4.1
Spongy cell: width, µm	8.3–23.1 14.7 ± 3.39	13.7–17.9 15.74 ± 1.57	7.4–21.5 14.2 ± 2.6
Abaxial epidermal cell: height, µm	11.2–24.3 19.6 ± 2.92	15–28 18.2 ± 3.33	7.7–18.1 13.5 ± 2.7

* The populations are represented by the number of the voucher specimens in SOA.

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
