# Peer review of "Crocus pallidus (Iridaceae)—A Neglected Species for the Bulgarian Flora and Critical Taxon in the Balkans"

_plants, 2022, doi:10.3390/plants11050686_

Round 1

Reviewer 1 Report

This paper aims to revise the taxonomic status and description of Crocus pallidus. Due to the lack of recent thorough taxonomic revisions and newly described species of the genus in the Balkan area, a detailed study like the one presented by the authors is most welcome and needed. The paper is generally clear and sound. My main concern is that, if I understood correctly, the authors have not studied the type material of C. pallidus at all. As well, they suggest that a recently described new species from Turkey (C. thracicus) is conspecific with C. pallidus and thus, a junior synonym of it. Again, they have apparently not investigated the type of that name. Although the current study can be considered sufficient to show that there is a neglected species of Crocus in Bulgaria, without studying the the original type material it cannot be confirmed that C. pallidus is the correct name for it  (although it may be a good guess). Because the aim of the paper is to clarify this situation, it is recommended to that the authors investigate the types to avoid any further confusion. Also, it would be great to get DNA material of C. thracicus to get better informed view of its status, although I don't consider this absolutely necessary in the current context. 

The phylogenetic part of the study needs some clarification. Which species in the tree actually belong to the section Nudiscapus? What was the basis for outgroup selection? Is the choice based on some earlier phylogeny? If so, please provide a reference and explain why this is suitable outgroup. I am not sure what the authors mean by "clearly developed clade" (line 197), but it does not sound phylogenetically correct terminology. It remains unclear to the reader what are the species from Aegean islands and Italy (line 201), since they are not named nor pointed out in the figure.

Other minor comments: 

Lines 18-19 there is something wrong in the sentence "The recently described as a new endemic species from Turkey..." , perhaps delete "as a"?

Line 21 the last sentence in the abstract is somewhat unclear, better to say that you suggest it should be treated as a synonym.

Table 2. Without being so familiar with Crocus anatomy, the first two characters (section width & section height) remain unclear. Please clarify what these measured section are. In relation to this, it would be nice to have the arms, keel zone and other key anatomical characters mentioned in the text pointed in the Fig. 5.

Line 308 C. palldus -> should be C. pallidus

Line 402 C. weldenii should be in italics

Lines 405-406: it remains unclear why the presence of hybrids between C. thracicus and C. chrysanthus is an indication that C. thracicus is closely related to C. pallidus?

Author Response

Thank you very much for the comprehensive review and the important notes.

Point 1:
This paper aims to revise the taxonomic status and description of Crocus pallidus. Due to the lack of recent thorough taxonomic revisions and newly described species of the genus in the Balkan area, a detailed study like the one presented by the authors is most welcome and needed. The paper is generally clear and sound. My main concern is that, if I understood correctly, the authors have not studied the type material of C. pallidus at all. As well, they suggest that a recently described new species from Turkey (C. thracicus) is conspecific with C. pallidus and thus, a junior synonym of it. Again, they have apparently not investigated the type of that name. Although the current study can be considered sufficient to show that there is a neglected species of Crocus in Bulgaria, without studying the the original type material it cannot be confirmed that C. pallidus is the correct name for it  (although it may be a good guess). Because the aim of the paper is to clarify this situation, it is recommended to that the authors investigate the types to avoid any further confusion. Also, it would be great to get DNA material of C. thracicus to get better informed view of its status, although I don't consider this absolutely necessary in the current context. 

Response 1:
The studied types are mentioned in Section 4.1 (page 13) of the manuscript, as follow:
C. pallidus:  SO 30393 (holotype), SO 30392, 32795 (isotypes).
C. thracicus: ISTE 102922 (holotype); DUOF 5630 (isotype).
Yes, we agree with the idea to study DNA from Turkish populations known as C. thracicus in future. 

Point 2:
The phylogenetic part of the study needs some clarification. Which species in the tree actually belong to the section Nudiscapus? What was the basis for outgroup selection? Is the choice based on some earlier phylogeny? If so, please provide a reference and explain why this is suitable outgroup. I am not sure what the authors mean by "clearly developed clade" (line 197), but it does not sound phylogenetically correct terminology. It remains unclear to the reader what are the species from Aegean islands and Italy (line 201), since they are not named nor pointed out in the figure.

Response 2:
The tree consists only of the representatives of the sect. Nudiscapus. We follow the approach in the cited publications, and the related topology of the published before phylogenetic trees (mentioned in the text). 
Following Harpke & al. (2017), we accept the C. adamii s.str. group as outgroup. 
In this case, we agree, that our outgroup in the tree was signed wrongly by us, and the tree could be rerouted. 
This mistake in the illustrated tree does not affect to the comments and the conclusions.
We hope the changes in Figure 7 could illustrate better the placement of C. pallidus.
For additional illustration we added the aligned sequences in Figure A1. We believe the next studies in Crocus with more ITS sequences could improve the quality of this phylogenetic tree.

Point 3:
Other minor comments

Response 3:
We did the corrections in the text. 

Reviewer 2 Report

The article is an interesting account on a scarcely known species of Crocus from the Balkans. Despite a lot of data presented and nice images, the conclusion are not well supported by the presented data.

Intro
pg 1 line 27: a taxonomic
line 7 C. chrysanthus in italics

The phylogenetic analysis is not very convincing.
Fig. 7 low quality: C. pallidus does not seem so different from the other belonging to Biflora.
The data support only the insertion in Biflora. 
The authors cite the non-monophyletic section Nudiscapus, that is not shown in the tree in Fig. 7.
The sentence: "The samples of C. pallidus are situated in the group of the poly- 293
morphic section Nudiscapus. It is forming a subclade in a clade with neighbors C. minutus 294
Kerndorff & Pasche [33] from Turkey (Lycia)." is not supported by the data (since C. pallidus is in a subclade that contains C minutus but also several other species) and however unclear.
I would say the last part of the discussion on phylogeny is not supported by fig. 7. Unless the authors are referring to another figure not provided.

Author Response

Thank you very much for the review and the important comments.

Point 1:
The phylogenetic analysis is not very convincing.
Fig. 7 low quality: C. pallidus does not seem so different from the other belonging to Biflora.
The data support only the insertion in Biflora. 
The authors cite the non-monophyletic section Nudiscapus, that is not shown in the tree in Fig. 7.
The sentence: "The samples of C. pallidus are situated in the group of the poly- 293
morphic section Nudiscapus. It is forming a subclade in a clade with neighbors C. minutus 294
Kerndorff & Pasche [33] from Turkey (Lycia)." is not supported by the data (since C. pallidus is in a subclade that contains C minutus but also several other species) and however unclear.
I would say the last part of the discussion on phylogeny is not supported by fig. 7. Unless the authors are referring to another figure not provided.

Response 1:
We agree with the critical note for the phylogenetic analysis.
Our idea is to follow the same approach as in previous studies, mainly the phylogenetic tree in Sect. Nudiscapus, published by Harpke & al. (2017).
The outgroup in Figure 7 was wrongly signed in the tree by our mistake. That's why we rerouted the tree in the manuscript, following the topology published before. The change in the figure does not affect the results and conclusions.
We hope the change in Figure 7 could explain our observations better.
We added the aligned sequences in Figure A1 for additional illustration.

Reviewer 3 Report

attached the manuscript with my observation.

Author Response

Thank you for the review, detailed notes, corrections and comments in the manuscript.

Point 1

Why do some relevant features of  C. weldenii lack in this table? Given the taxonomic position of C. pallidus under C. weldenii a comparison with data from floras and herbarium specimens (which can recovery on line) should have been more appropriate

Response 1

Yes, we agree with your notes about Table 1. We added some characteristics taken from herbarium sheets available via GBIF. We also added descriptions of the specimens in Sect. 4.1 (page 13).

Round 2

Reviewer 1 Report

I think the paper is ok except for some phylogenetic discussions. Crocus pallidus is not a "neighbour" of C. minutus (there are no such thing as "neighbours" in phylogenetics) – the phylogenetic positions of these species are simply unresolved and they just happen to be placed next to each other in the tree, but the species with unresolved relationships in the clade could as well be arranged in any other order. Also, C. alexandri and C. weldenii are not a sister clade to a "clade" of C. punctatus, C. leucostylus and C. randjeloviciorum. The latter three species do not for a clade at all, an the phylogenetic relationships between these species, as well as the relationship between them and the C. alexandri + C. weldenii clade are unresolved. Based on the presented data it is not possible to say how these species are related to each other, except that they all belong to the same larger clade. As well, "outstanding" is not a proper term to describe non-monophyly. These incorrect interpretations of the tree must be corrected.

Author Response

Thank You again for your review

Point 1

I think the paper is ok except for some phylogenetic discussions. Crocus pallidus is not a "neighbour" of C. minutus (there are no such thing as "neighbours" in phylogenetics) – the phylogenetic positions of these species are simply unresolved and they just happen to be placed next to each other in the tree, but the species with unresolved relationships in the clade could as well be arranged in any other order. Also, C. alexandri and C. weldenii are not a sister clade to a "clade" of C. punctatus, C. leucostylus and C. randjeloviciorum. The latter three species do not for a clade at all, an the phylogenetic relationships between these species, as well as the relationship between them and the C. alexandri + C. weldenii clade are unresolved. Based on the presented data it is not possible to say how these species are related to each other, except that they all belong to the same larger clade. As well, "outstanding" is not a proper term to describe non-monophyly. These incorrect interpretations of the tree must be corrected.

Response 1

Yes, we agree with your notes. It is enough to mention the position of C. pallidus in the large clade of ser. Biflori. In this case, we removed the incorrect comments.

Reviewer 2 Report

the current version with the modification of fig. 7 answer my previous concerns

Author Response

Thank You very much.

We uploaded a new version of the manuscript according to the note of the other reviewer.
